# A Narrative Review of Sulfur Compounds in Whisk(e)y

**DOI:** 10.3390/molecules27051672

**Published:** 2022-03-03

**Authors:** Akira Wanikawa, Toshikazu Sugimoto

**Affiliations:** Technical Development Center, The Nikka Whisky Distilling Co., Ltd., 967 Masuo, Kashiwa, Chiba 2770033, Japan; toshikazu.sugimoto@nikkawhisky.co.jp

**Keywords:** malting, fermentation, distillation, maturation, sensory analysis, chemical analysis

## Abstract

The production process of whisky consists of malting, mashing, fermentation, distillation and maturation. Sulfur volatile compounds generated during this process have long attracted interest because they influence quality in general. More than forty compounds have been reported: they are formed during malting, fermentation, and distillation, but some may decrease in concentration during distillation and maturation. In sensory analysis, sulfur characteristics are described as sulfury, meaty, cereal, feinty, and vegetable, among others. Their contribution to overall quality depends on their concentration, with a positive contribution at low levels, but a negative contribution at high levels. Chemical analyses of sulfur volatiles have been developed by using sulfur-selective detectors and multi-dimensional gas chromatography to overcome the numerous interferences from the matrix. Formation pathways, thresholds, and contribution have not been elucidated completely; therefore, methods for integrating diverse data and knowledge, as well as novel technical innovations, will be needed to control sulfur volatiles in the future.

## 1. Introduction

Whisky, a distilled spirit that is consumed worldwide, is produced from grains such as barley, corn, wheat, and rye by a process of malting, mashing, fermentation, distillation, and maturation. There are five major types of whisky in the world, Scotch, Irish, American, Canadian, and Japanese (Table 1) [1,2,3,4,5,6,7], which differ in the raw materials and production processes, especially the distillation method. There are two types of Scotch whisky, malt and grain whisky, which are made from malted barley and wheat and/or corn plus approximately 10% malted barley, respectively; the malt whisky is distilled twice using a copper batch still, while grain whisky is produced using a continuous still. Irish whiskey, the production process of which is similar to that of Scotch whisky, is produced from malted barley and barley, or corn and wheat plus malted barley or commercial enzymes for the decomposition of starch. By contrast, American and Canadian whiskey is produced from corn, wheat, or rye, and malt and/or enzymes, with the spirit produced by continuous distillation. Lastly, Japanese whisky, including malt and grain whisky, is produced in the same style as Scotch whisky, based on Scottish technologies and knowledge introduced 100 years ago.

During whisky production, starch and proteins are degraded to sugars and amino acids mainly by malt or commercial enzymes during the mashing process; ethanol is produced from the sugars by yeast during fermentation; the ethanol is concentrated during distillation; and the final spirit is matured in oak. Various flavor compounds are formed and also reduced in quantity during these processes [8,9], as shown in Figure 1. Sulfur, heterocyclic, and phenolic compounds are formed during malting. In the fermentation process, yeast is involved in the formation of several volatile compounds, including esters, fusel alcohols, sulfur compounds, lactones, and aldehydes as by-products, alongside the production of ethanol. Ketones are produced during distillation; moreover, sulfur compounds are formed and reduced in this process. Lastly, numerous flavor compounds such as lactones, aromatic phenols, heterocyclic compounds, and tannins are extracted during oak maturation. In this process, the number and concentration of sulfur volatile compounds must decrease because these have unpleasant, ultimately undesirable characteristics.

Considering in more detail the formation of flavors in each process, the malted barley is derived from the malting process, which involves steeping, germination, and kilning. Several enzymes are formed during germination, especially α- and β-amylase for the degradation of starch, and carboxypeptidases for the supplementation of amino acids, which play an important role in yeast fermentation [10]. Sometimes peat, an organic sediment derived from grass, moss, or heath, is used for kilning in Scotch and Japanese whisky-making and provides phenolic compounds including alkylphenols and methoxy alkyl phenols [8,9,10]. These compounds are known to give a peaty character; however, this character is likely to be complex, owing to the multitude of compounds. Steele et al. pointed out that, in sensory evaluations, the complex phenolic character consists of medicinal, smokey, and burnt characteristics [11].

The next process is mashing, during which starch is converted into fermentable sugars. There are two types of mashing: separation from solid, and whole mash [1,4,10]. In Scotch and Japanese malt whisky making, the malt is milled and mixed with hot water, and the starch is converted into fermentable sugars by enzymes in the malt. Normally, the mashing takes place at 63 °C for at least 1 h [10]. Residual sugars are recovered by sprinkling in hot water two or three times after the first wort is run-off. In this respect, malt whisky could be classified as a distillate from unhopped beer because the materials and processes in whisky-making are as also used in brewing. For American and Canadian grain whiskey, by contrast, grains are milled, mixed with hot water and/or enzymes, and heated at approximately 70 °C for gelatinization (the temperature varies for different types of grains [12]. Again, the natural enzymes from malt or commercial enzymes are added to convert starch to carbohydrates that yeast can metabolize to ethanol. There are few reports on the formation of flavors in the mashing of whisky; however, this process is similar in brewing and whisky making, and breweries have reported the formation of off-flavors such as aldehydes due to the oxidation of lipids [13,14].

After the wort or mash is cooled to around 20 °C, yeast (*Saccharomyces cerevisiae*) is added. The fermentation period in spirits is likely to be 2–5 days, during which yeast forms ethanol and carbon dioxide, along with volatiles such as esters, fusel alcohols, and sulfur compounds [15,16,17]. For example, ethyl hexanoate and 3-metylbutyl acetate yield fruity characters, and 2-phenylethyl acetate gives a floral one. Although differences might be expected due to the region and raw materials used, almost all fermentable sugars are converted to ethanol and the ethanol concentration reaches 7–10% alcohol by volume (ABV). Because solids are not separated before fermentation and a continuous still is used in the production of American whiskey (bourbon), the final product contains higher amounts of fusel alcohols and lower amounts of esters, as compared with Scotch malt whisky [18].

After fermentation, distillation is carried out in either a batch still or a continuous still, and ethanol along with volatiles is concentrated during this process. The ethanol concentration depends on the production country’s regulations and the type of distillation. For example, it is 65–70% ABV in Scotch and Japanese malt whisky, but less than ~95% ABV in grain whisky. A typical volatile formed during distillation is (*E*)-1-(2,6,6-trimethyl-1-cyclohexa-1,3-dienyl)-but-2-en-1-one (β-damascenone), produced via the hydrolysis of glycosides due to heating [19]. This compound yields flowery and ionone-like characters, and may be present at more than 10 times the threshold in Scotch and Japanese whiskies. In addition, it has been reported that the surface of a copper still removes sulfur compounds, while other studies indicated the formation of alkyl sulfides during distillation.

In the last step, distilled spirits are diluted with water to 60–70% ABV and transferred to oak wood barrels. Scotch malt whisky should be matured for 3 years at least, while straight bourbon and other American straight whiskeys are aged for a minimum of 2 years in new charred oak [10]. The maturation process has three roles: the extraction of oak flavor, the adsorption of unpleasant flavor, and mellowing [9,20]. Regarding the first role, it is well known that sweet flavors are extracted from oak, such as vanilla-like, coconut-like, and clove-like aromas, and a sweet taste, coupled with color development [10,20,21,22,23,24]. Typical examples of these compounds include 3-methoxy-4-hydroxybenzaldehyde (vanillin) and 3-methyl-4-octanolides, known as “oak lactones” or “whisky lactones”. In addition, taste compounds such as ellagitannins, triterpenoids, and lignans have been reported. The developing brown color is likely to be due to polyphenol compounds; however, the components have not been fully elucidated.

Regarding the second role, the inside of oak is charred, and the carbonization layer can remove unpleasant sulfur compounds, including alkyl sulfides such as methylsulfanylmethane (dimethyl sulfide, DMS), methyldisufanylmethane (dimethyl disulfide, DMDS), and methyl trisulfanylmethane (dimethyl trisulfide, DMTS). Moreover, these compounds are also likely to evaporate over time during maturation. Lastly, maturation over time results in a reduction in alcoholic pungency. This phenomenon is thought to be due to the formation of clusters between ethanol and water molecules. Recently, it was reported that two types of clusters, small and large, form during maturation, and it is the large ones that contribute to reducing alcoholic pungency [25].

Due to the complex matrix of whisky, the industry has relied on sensory evaluations for a long time; therefore, the relationships between flavor and constituents have not been completely elucidated. Regarding malt whisky, in 1979 a Scotch whisky research group published a flavor wheel [26], a tool to help sensory descriptions based on a graphical representation of the systematic categorization of all attributes. Since then, other researchers produced revised flavor wheels [27,28] including reference compounds to provide systematic knowledge and understanding for quality and process, and to facilitate communication among blenders, distillers, and researchers toward the improvement of production processes. Therefore, these latter wheels might be regarded as professionally oriented ones, while those for American whiskey are likely to be consumer-oriented and proposed by distributors and distillers [29] (pp. 56–58). In short, the flavor wheels of Scotch and American whisky differ greatly on several points, including concept and target audience.

The whisky industry has believed for a long time that the sulfur volatile compounds that are derived from malt and formed during fermentation generally have undesirable characteristics, decrease with copper during the distillation process, and evaporate or are adsorbed by oak wood in the maturation process. However, low levels of sulfur volatile compounds contribute positively to quality, providing full body and complexity, while high levels contribute negatively [14]. Therefore, the contribution of sulfur compounds to quality and attributes, including thresholds, remains to be fully revealed.

This mini-review focuses on sulfur compounds in whisky. First, the formation of and decrease in different compounds in each process are discussed; subsequently, chemical and sensory analyses are described. In addition, chemical analyses used for other spirits are briefly considered because they might be useful and inform the whisky industry. Finally, the control of sulfur volatiles, for example, by using active carbon and filtration technologies, is described.

## 2. Formation and Removal of Sulfur Compounds in the Whisky Production Process

As described above, whisky is produced through malting, mashing, fermentation, distillation, and maturation. During these processes, several sulfur compounds are formed and/or reduced in quantity, as shown in Figure 1. To date, alkyl sulfides [30,31,32,33,34,35], thiols [30], methylthio group compounds [31,35], thiophenes [31,35,36], thiazoles [31,35,36,37,38], dithiapentane derivatives [33], and furfuryl compounds [34] have been detected in the final spirit. Distillers might be able to control quality better if the pathways and contributions of these chemicals were known; currently, however, only some of them have been identified, while others remain to be revealed. In this section, we describe how some of these compounds are formed, as well as where their levels are decreased, in the whisky-making processes.

### 2.1. Formation of Dimethyl Sulfide in Malting

In the malting process, barley is initially steeped in water and then rested in air for 36–48 h at 16 °C. Subsequently, the water is drained off and the barley is allowed to germinate for around 5–6 days. The germinated barley is then kilned at 60–70 °C for 24 h [10].

There have been several reports on the formation of DMS during malting. DMS is described as “cooked sweet corn”, “cooked vegetables”, and “cooked tomato” [39,40], and its threshold is reported as 5 μg/L in 20% ethanol solution [41]. According to Bamforth [42], *S*-methyl methionine is formed from methionine during germination and converted to DMS during kilning, and at higher kilning temperatures, the amount of DMS will be increased. Bathgate also claimed that both the kilning temperature and the design of the equipment affect the character of the malt, including sulfur and vegetable-like aromas [43]. Therefore, temperature might be a key target for decreasing the amount of DMS that forms. In addition, during kilning, a portion of DMS might oxidize into odorless methylsulfonylmethane (dimethyl sulfoxide), which would then be converted back to DMS by yeast during fermentation [39].

### 2.2. Formation of Sulfur Donors during Fermentation

After mashing using malt or commercial enzymes, yeast is added to the cooled wort or mash. During fermentation, yeast converts sugars into ethanol and carbon dioxide, but is also involved in the formation of sulfur compounds. Miller summarized the sulfur compounds found in wort and whisky as follows: three alkyl sulfides, DMS, DMDS and DMTS; three thiols, sulfane, methanethiol and ethanethiol [29] (p171); five methylthio derivatives, such as 3-methylsulfanylpropane-1-ol (3-(methylthio) propanol, MTP); and seven thiophene derivatives, including thiophene, 2-methylthiolan-3-one, 2-methylthiophene, benzothiophene, benzothiazole, and 2-methyl-3-(methyldisulfanyl)furan (methyl-(2-methyl-3-furyl) disulfide, MMFDS) (Table 2 and Figure 2).

**Table 2 molecules-27-01672-t002:** Sulfur volatile compounds found in whisky. (a) The Good Scents Company Information System (http://www.thegoodscentscompany.com/, accessed on 23 February 2022). nd, not described on the site. (b) Tentative identification.

No.	Class	Compound Name	Common Name	Odor Description ^(^^a)^	References
1	sulfides	methylsulfanylmethane	dimethyl sulfide	sweet corn	Leppänen et al. [44]
2		ethylsulfanylethane	diethyl sulfide	garlic-like	Masuda and Nishmura [31]
3		1-propylsulfanylpropane	dipropyl sulfide	garlic, onion	Campillo et al. [32]
4		methylsulfanylpropane	methyl propyl sulfide	green, leek	Campillo et al. [32]
5		methyldisulfanylmethane	dimethyl disulfide	vegetable	Leppänen et al. [44]
6		propyldisulfanylpropane	dipropyl disulfide	green onion	Campillo et al. [32]
7		2-methyl-1-(methyldisulfanyl)propane	iso-butyl methyl disulfide	nd	MacNamara [35] ^(b)^
8		methyltrisulfanylmethane	dimethyl trisulfide	onion, meaty	Leppänen et al. [44]
9		2-methylsulfanylethanol	2-(methylthio) ethanol	meaty	Taniguchi et al. [33]
10		3-methylsulfanylpropan-1-ol	3-(methylthio) propanol	boiled potato	Masuda and Nishimura [31]
11		3-methylsulfanylpropanal	3-(methylthio) propanal	onion, meaty	Masuda and Nishimura [31]
12		3-methylsulfanylpropyl acetate	3-(methylthio)propyl acetate	potato	Masuda and Nishimura [31]
13		*S*-methyl ethanethioate	*S*-methyl thioacetate	cheese	Leppänen et al. [44]
14		2-(methyldisulfanyl)ethan-1-ol	3,4-dithiapentyl alcohol	nd	Taniguchi et al. [33]
15		1-ethoxy-2-(methyldisulfanyl)ethane	3,4-dithiapentyl ethyl ether	nd	Taniguchi et al. [33]
16		2-(methyldisulfanyl)ethyl acetate	3,4-dithiapentyl acetate	nd	Taniguchi et al. [33]
17		2-methyl-3-(methyldisulfanyl)furan	methyl-(2-methyl-3-furyl) disulfide	meaty, sulfury	Cater-Tjimstra [34]
18	thiols	sulfane	hydrogen sulfide	rotten egg	Ronkainen et al. [30]
19		methanethiol		rotten cabbage	Ronkainen et al. [30]
20		ethanethiol		leek	Ronkainen et al. [30]
21	mercapto esters	ethyl 3-methylsulfanyl propanoate	ethyl 3-(methylthio) propanoate	pineapple	Masuda and Nishimura [31]
22		ethyl 2-methylsulfanyl acetate	ethyl 2-(methylthio) acetate	green tropical	MacNamara [35]
23	thiophenes	thiophene		garlic	Masuda and Nishimura [31]
24		2-methylthiophenone		meaty, cooked	Masuda and Nishimura [31]
25		2,5-dimethylthiophene		nutty, green	Masuda and Nishimura [31]
26		thiophene-2-carbaldehyde	thiophene-2-carboxaldehyde	benzaldehyde-like	Masuda and Nishimura [31]
27		thiophene-3-carbaldehyde	thiophene-3-carboxaldehyde	nd	Ochiai et al. [36]
28		3-methylthiophene-2-carbaldehyde	3-methylthiophene-2-carboxaldehyde	nd	Ochiai et al. [36]
29		3-ethylthiophene-2-carbaldehyde	3-ethylthiophene-2-carboxaldehyde	nd	Ochiai et al. [36]
30		5-methylthiophene-2-carbaldehyde	5-methylthiophene-2-carboxaldehyde	benzaldehyde-like	Masuda and Nishimura [31]
31		2-methylthiolan-3-one	dihydro-2-methyl-3(2H)-thiophenone	sulfur, fruity	Masuda and Nishimura [31]
32		1-thiophen-2-ylethanone	2-acetyl thiophene	nutty	MacNamara [35]
33		1-thiophen-2-ylbutan-1-one	2-butanoyl thiophene	meaty	MacNamara [35]
34		1-(5-methylthiophen-2-yl)ethanone	2-acetyl-5-methyl thiophene	sweet, spicy	MacNamara [35] ^(b)^
35		1-benzothiophene		rubbery	Masuda and Nishimura [31]
36	thiazoles	1,3-thiazole		nutty, meaty	Masuda and Nishimura [31]
37		2-methyl-1,3-thiazole		vegetable	Ochiai et al. [36]
38		1-(1,3-thiazol-2-yl)ethanone	2-acetyl-1,3-thiazole	popcorn	MacNamara and Hoffmann [38]
39		5-ethenyl-4-methyl-1,3-thiazole	4-methyl-5-vinyl-1,3-thiazole	nutty	Piggott [37]
40		1,3-benzothiazole		rubbery	Masuda and Nishimura [31]
41		2-methyl-1,3-benzothiazole		rubbery, coffee	Ochiai et al. [36]
42		3-ethyl-1,3-benzothiazole-2-thione	3-ethyl-1,3-benzothiazolethione	nd	Ochiai et al. [36]
43		2-(furan-2-yl)-1,3-thiazole	2-(2-furanyl)-thiazole	nd	MacNamara [35]

Masuda and Komura also reviewed the formation of flavors including sulfur compounds during fermentation [18]. According to them, MTP, 3-methylsulfanylpropyl acetate (3-(methylthio) propyl acetate, MTPA), and 2-methylthiolan-3-one were formed from methionine during fermentation in malt whisky production. Schreier et al. examined metabolites formed from methionine as the sole nitrogen source, reporting the formation of MTP at 50–60% and trace amounts of MTPA, 3-methylsulfanyl propanal, and 2-methylthiolan-3-one [45]. More recently, Etschmann et al. studied the formation of MTP and MTPA from methionine by *S. cerevisiae* [46]. While the wild-type yeast produced large amounts of MTP and trace MTPA in a synthetic medium containing methionine as the sole nitrogen source, considerable amounts of both MTP and MTPA were produced by a genetically modified yeast. Thus, MTP and MTPA seem to be formed from methionine by yeast during fermentation; these compounds have onion-/potato-like, and sulfurous characteristics, respectively. Deed et al. also studied the formation of MTA from methionine by *S. cerevisiae* [47]. They examined three Ehrlich pathway genes, which suggested that alternative pathways may be involved in the formation.

Hydrogen sulfide is known to be unpleasant, with a rotten egg aroma, and can act as a sulfur donor due to its high reactivity. Stewart and Ryder discussed the sulfur cascade during fermentation, especially from sulfate to sulfur-containing amino acids [39]. It is well established that *S. cerevisiae* can assimilate sulfate into cells, incorporating sulfur dioxide and hydrogen sulfide into the amino acids cysteine and serine via the *MET17* gene in the methionine biosynthesis pathway. A portion of sulfur dioxide and hydrogen sulfide may also leak out of yeast cells. In biochemical and molecular biological studies using the stable isotope ^34^S and a mutant yeast, Kinzurik et al. showed that hydrogen sulfide could be converted into ethanethiol, *S*-ethyl thioacetate, and diethyl disulfide [48]. Because the yeast strain with *MET17* gene deletion ultimately accumulated hydrogen sulfide, it might be possible to identify metabolites of hydrogen sulfide. Moreover, the same research group proposed the conversion of hydrogen sulfide [49], ethanethiol, and methanethiol both through biological transformation by *S. cerevisiae* and through chemical conversion. Thus, hydrogen sulfide might be converted to ethanethiol by yeast, and ethanethiol might be converted to ethylsulfanylethane via *S*-ethyl thioacetate. In addition, hydrogen sulfide and ethanethiol might be turned into methanethiol chemically. From the methanethiol, three alky sulfides (DMS, DMDS, and DMTS) might be produced by chemical reaction, while *S*-methyl thioacetate might be formed by yeast. It seems, therefore, that hydrogen sulfide, ethanethiol, and methanethiol may act as sulfur donors, forming several sulfur volatiles during the fermentation process, although the kinetics and equilibria of the reaction remain to be elucidated. Ethanethiol and ethylsulfanylethane have garlic-like aromas, while methanethiol has a rotten cabbage character. The threshold of ethanethiol has been reported as 0.03 μg/L in 20% ethanol [41]. As a side note, hydrogen sulfide has also been reported to act as a sulfur donor in the formation of an onion-like off-flavor in beer, namely 2-methyl-3-sulfanylbutan-1-ol [50,51]. Hence, because the formation of hydrogen sulfide is associated with fermentation speed and sulfur-containing amino acids, fermentation conditions might be a key target for controlling unpleasant sulfur volatiles. Investigations into yeast strain, fermentation temperature, and yeast inoculation, among other factors, might lead to better control of these compounds in the future.

Hydrogen sulfide and other sulfur volatiles that are formed from hydrogen sulfide might be associated with the sulfurous character reported in the following two studies. First, Daute et al. examined the influence of pretreatment of wort [52], including boiling, autoclaving, and filtration, on sensory and chemical analysis, demonstrating that larger amounts of volatiles, such as esters, fusel alcohols and sulfides, were formed as compared with control wort, although no significant differences were detected between pretreated and control wort. However, pretreated wort had more feinty characteristics, such as meaty, cereal, and sulfury notes. It was suggested that these changes might be due to differences in the components of wort, because there was a decrease in fermentation speed and ethanol yield. Second, Waymark and Hill investigated the influence of 24 yeast strains [53], including distilling yeasts, wine yeasts, and brewer’s yeasts, in malt whisky fermentation, carrying out sensory analysis on new-make spirits prepared on the laboratory scale. They demonstrated a difference in sensory intensity among the new-make spirits; in particular, brewer’s yeasts produced whisky with higher scores of cereal, sulfury, and feinty notes.

Sulfur volatiles produced by brewer’s yeast have also been reported. Yomo et al. studied the effect of brewer’s yeast [54], which was originally used in malt whisky making many years ago. Indeed, some distilleries use a mixture of distilling and brewer’s yeast even today, although most distilleries use only distilling yeast. The researchers demonstrated that new-make spirits produced with the mixed yeasts had a full-body character and increased amounts of sulfur compounds, including DMTS, 2-(methyldisulfanyl) ethanol, and 2-(methyldisulfanyl) ethyl acetate. The dithiapentyl derivative was reported to have a mushroom character. The researchers also compared fresh brewer’s yeast with starved yeast, showing that the latter produces spirits with a more complex and full-body character in sensory analysis due to the formation of larger amounts of these sulfur compounds. They suggested that physiological changes in the starved yeast lead to desirable product qualities after maturation. Overall, their findings are identical to those of Waymark and Hill [53], who showed that brewer’s yeast tends to produce a more intense feinty note, although there is a difference between fresh and spent yeast.

### 2.3. Formation and Removal of Sulfur Volatiles during Distillation

The behaviors of sulfur volatile compounds during distillation have been previously explored in relation to the copper still. Normally, malt whisky is distilled twice: a first distillation from wash to low wine, and a second distillation from low wine to spirits. During distillation, water is supplied to the condenser, which condenses the vapor back to liquid. There are two types of condenser: a traditional worm tub, composed of a coil in a tub; and a shell-and-tube, composed of numerous small tubes in big shells [55]. It is generally believed that the type of condenser affects the product quality, especially sulfur characters, with the worm tub producing whisky with greater sulfury and heavier notes due to the surface area and duration of contact with copper, as Bathgate has pointed out [43].

The level of copper in the distillate is an indicator of the consistency of spirits, and higher levels are identified in those produced by a shell-and-tube condenser [12], which is why this type of condenser provides a lighter quality. For differentiation and authentication, Hopfer et al. determined 53 elements in multiple whiskies from different countries [56], reporting higher levels of copper in Scotch and Japanese whisky than in Irish, bourbon, and Tennessee whiskeys, although they did not discuss the reasons. Webster et al. evaluated the influence of the two types of condenser on the amounts of alkyl sulfides, MMFDS, and copper in spirits from two commercial distilleries [57]. They observed that the amount of MMFDS in whisky from the distillery with the worm tub condenser was lower than that from the distillery with the shell-and-tube condenser. However, the operation and shape of the pot still also differed between the two distilleries, and the researchers concluded that other factors, in addition to condenser type, might affect the behavior of sulfur volatiles during distillation.

Distillation is reported to cause both the formation of sulfur volatiles and a decrease in sulfur levels, leading Miller to suggest that it might be necessary to integrate the two theories and propose new mechanisms [29] (p. 172). Based on their experience, distillers believe that copper removes sulfur compounds, as mentioned above; thus, a decrease in levels might be considered to occur during whisky production. On the contrary, however, hydrogen sulfide and alkyl thiols—the proposed precursors of DMDS and DMTS—may react easily with other compounds in the presence of copper, as described in the fermentation section. Therefore, it seems reasonable that both reactions might occur during distillation, although the mass balance of the two will vary. The threshold of DMDS and DMTS has been reported as 30 and 0.05 μg/L in 20% ethanol solution, respectively [41].

Regarding the formation of sulfur compounds, Masuda and Nishimura compared amounts of DMDS in whisky prepared using copper and glass stills of the same shape [31], demonstrating that the amount in the distillate from the glass still was about one-tenth that of the copper still. Nedjma and Hoffmann studied the formation of alkyl sulfides from hydrogen sulfide, methanethiol, and ethanethiol in a buffer solution in the presence of Cu^2+^ [58]. They proposed that DMDS was formed from methanethiol, while DMTS was formed from methanethiol and hydrogen sulfide. Furusawa proposed that DMDS and DMTS were formed from MTP and from MTP with hydrogen sulfide during distillation owing to copper salts [59]. All in all, while the last two studies were performed in model solutions, it will be clearly difficult to prove these reactions in distillation.

In terms of a decrease in the amounts of sulfur compounds, Jack et al. evaluated the effects of copper and non-copper stills on sulfury sensory scores, such as cereal, feinty, sulfury, and meaty notes [60]. They carried out three distillation tests using all copper, non-copper in the first and copper in the second distillation, and the reverse combination (copper and non-copper). They observed that all-copper and copper/non-copper distillation led to lower sensory scores, while non-copper/copper had significantly higher scores, suggesting that the contact wash with copper led to a decrease in sulfur compounds. Subsequently, Harrison et al. examined the importance of copper in more detail [14]. They designed a still that could be interchanged with copper or stainless steel in three parts: the pot, lyne arm, and condenser. Distillation experiments were carried out on a laboratory scale, the amount of DMTS in the distillates was determined by using a headspace GC–sulfur chemiluminescence detector (SCD), and the score in terms of meaty and sulfury notes in sensory analysis was evaluated. In short, the authors attempted to identify crucial contact points—namely, liquid in the pot, vapor in the lyne arm, or liquid in the condenser. They demonstrated that both the amount of DMTS and the sensory score were lower in all-copper distillation. In terms of the three still parts, the pot had the most effect, although not all 64 combinations were examined. This study indicated that contact with copper for both liquid in the pot and vapor in the lyne arm is the most important for reducing levels of sulfur compounds, while copper salt was involved in their formation. In addition, two unknown peaks were observed in GC-SCD analysis, especially in distillate from the stainless steel still, suggesting that these compounds might contribute to meaty and sulfury characteristics.

### 2.4. Decrease in Alkyl Sulfides during Maturation

Distilled spirits are matured in oak casks. In regard to sulfur volatiles, there have been a few novel findings in recent years. It is known that the rate of decrease in sulfur compounds during maturation depends upon the sulfur compounds. Masuda and Nishimura measured sulfur volatiles, including three alkyl sulfides [31], three thiophenes, four MTP derivatives, and benzothiazole, in malt whisky that was unaged and aged for 1–10 years. There was observed three types of behavior: DMTS, 5-methylthiophene-2-carbaldehyde, benzothiophene, and benzothiazole did not change; DMDS gradually decreased; and other compounds decreased markedly within a few years. In another study, Leppänen et al. reported that DMTS decreased slowly, consistent with currently held views [44,61]. Thus, DMS decreases markedly, DMDS decreases gradually, and DMTS slowly decreases due to the charcoal layer inside the oak. As pointed out by Masuda and Komura [18], these alkyl sulfides contribute to typical immature characteristics and their presence indicates the lack of a substantial maturation period.

## 3. Chemical Analysis of Sulfur Compounds

Sulfur volatile compounds that have been reported in whisky are summarized in Table 2. Sulfides, thiols, mercapto esters, thiophenes, and thiazoles were identified in whisky 50 years ago. Because larger amounts of other volatiles, such as ethanol (percent order), fusel alcohols (hundred ppm order) and esters (ppm order), are present relative to sulfur volatiles (ppt to hundred ppb), sample preparation before analysis by GC, sulfur-selective detectors [62] such as a flame photometric detector (FPD), SCD, and multi-dimensional GC [61,62,63,64,65] might be helpful in the analysis of such minor volatiles. The analytical methods used to date are summarized in Table 3.

**Table 3 molecules-27-01672-t003:** Analytical procedures of sulfur volatile compounds in whisky.

Sample Preparation	Separation	Detector	References
headspace	GC	FPD	Ronkaine et al. [30]
headspace	GC	FPD	Leppänen et al. [66]
liquid/liquid extraction	GC	MS	Masuda and Nishimura [31]
liquid/liquid extraction	MDGC	MS, ECD	Carter-Tijmstra [34]
not shown	MDGC	SCD, NTD, MS	MacNamara an Hoffmann [38]
vacuum distillation, preparative GC	MDGC	SCD, MS	MacNamara [35]
counter-current chromatography	GC	MS	Taniguchi et al. [33]
headspace SPME	GC	MS	Campillo et al. [32]
headspace Tenax	GC	SCD	Harrison et al. [14]
full evaporation dynamic headspace	MDGC	SCD, NCD, MS	Ochiai et al. [36]
headspace SPME	GC	MS	Dziekońska-Kubczak et al. [67]
solvent assisted flavor extraction	GC	MS	Kerley and Munafo [68]
headspace SPME	GC	MS	Daute et al. [13]

Carter-Tijmstra initially detected MMFDS in grain whisky by multi-dimensional GC coupled using an electron capture detector (ECD) and a heart-cutting technique [34], and pointed out its presence also in malt whisky. Since then, Watt et al. demonstrated that half of the new-make malt whiskies that they studied contained a considerable amount of MMFDS, more than the threshold (0.015 μg/L in 20% ethanol solution) [41]; they suggested that this compound might contribute to the meaty character in malt whisky, although not all samples possessed a meaty character.

Alkyl sulfides such as DMS, DMDS and DMTS have been considered markers of maturation. Ronkainen et al. first detected these alkyl sulfides in a grain spirit [69], and subsequently detected them in 13 blended and 3 malt whiskies [61,66], reporting that DMDS decreased rapidly during maturation, while DMTS decreased slowly, by comparison between standard and more aged whiskies. They prepared samples under nitrogen gas purge and used an absorbent trap to measure the amounts using GC-FPD. Masuda and Nishimura also identified several sulfur volatile compounds in malt whisky, including six sulfides, one mercapto ester, three thiophenes, and two thiazoles by GC-FPD and GC-mass spectrometry (MS) using a liquid extraction method [31].

With regard to sample preparation, Taniguchi et al. identified three dithiapentyl derivatives, namely, 2-(methyldisulfanyl) ethanol, 2-(methyldisulfanyl) ethyl acetate and 1-ethoxy-2-(methyldisulfanyl) ethane, which had a similar mushroom-like character at low levels [33]. They fractionated the volatiles by using centrifugal counter-current chromatography. Other sulfur compounds, including DMTS, MTP, 2-methysulfanylethnol, 2-methylthiolan-3-one, thiophene-2-carbaldehyde, and 1-(1,3-thiazol-2-yl) ethenone, were also detected by this method.

Using a multi-column switching method coupled with MS, SCD, and a nitrogen thermionic detector (NTD), MacNamara identified 2-methylthiolan-3-one, thiophene-2-carbaldehyde, 1-(1,3-thiazol-2-yl) ethanone, and MTPA in whisky [38]. Campillo et al. reported a headspace solid-phase microextraction (SPME) method using GC coupled with an atomic emission detector for the quantification of eight alkyl sulfides [32], including DMS, methylsulfanylpropane, and DMDS. Furthermore, Ochiai et al. simultaneously quantified 20 sulfur compounds using a full evaporation dynamic headspace method, coupled with a a heart cutting technique, MS and SCD [36], identifying seven sulfur volatiles in whisky for the first time. In addition, they detected eight unknown sulfur compounds in a new-make malt whisky and identified 1-ethoxy-2-(methyldisulfanyl) ethane by SCD, MS and the calculated formula. Recently, Dziekońska-Kubczak et al. proposed a method for the determination of sulfur volatiles in fruit brandy using headspace SPME/GC-MS [67]. They optimized extraction conditions, including extraction duration, ethanol concentration, and the addition of NaCl and EDTA. This approach might be useful for the analysis of other spirits, including whisky, rum and tequila.

In summary, chemical analysis of sulfur volatiles in whisky has been performed with sulfur-selective detectors and multi-dimensional GC in order to quantify significant minor components in this complex matrix. Because the development of analytical techniques has been marked in recent years, further investigations of sulfur volatiles in whisky are expected in the near future.

## 4. Sensory Evaluation and Its Contribution to Quality

Although more than 40 sulfur volatiles have been identified, the following questions remain. Which compounds contribute to the quality and to sulfury attributes such as feinty, cereal, and meaty notes? Which compounds are present at levels above their threshold?

The flavor wheel has become a foundation in sensory evaluation. The first wheel was created in 1979 by a research group consisting of blenders and scientists [26]. The terminology was constructed by summarizing and sorting descriptions, providing definitions including a choice of standard samples and reference compounds, and by the specification of overall impressions, with a wide input of ideas from industry. The first wheel consisted of 14 first tiers, 12 aromas, 1 taste, and 1 mouthfeel attribute. Among these, the sulfur attribute was defined in the first tier, and included four second tiers: stagnant, coal-grass, rubbery, and cabbage water, the last two of which were new. DMS was included in the third tier. It was considered that the wheel provided useful information and facilitated communication within the industry.

Subsequently, Lee et al. produced a flavor wheel and reference compounds, consisting of 13 aromas and 3 taste attributes in the first tier [27]. According to their wheel, the sulfury attribute consisted of six second tiers: stagnant, meaty, vegetable, sour, gassy, and rubbery. The reference compounds and levels were included in the third tier; for example, stagnant, meaty, vegetable, and gassy were referred to as DMTS at 3 ppm, MMFDS and DMS each at more than 0.6 ppm, and ethanethiol at more than 0.072 ppm, respectively. Jack et al. reported a wheel in which sulfury comprised five attributes, namely, cooked vegetable, rubbery, struck match, decaying, and meaty [28]. They also used 15 attributes for new-make spirits: pungent, phenolic, feinty, cereal, floral, aldehydic, estery, solvent, oily, sour, soapy, sulfury, meaty, stale, and clean [70]. Of these, sulfur volatiles are likely to contribute to the four attributes feinty, cereal, sulfury, and meaty. Overall, their wheel and evaluation seem to be similar to that of Lee et al., although reference compounds and definitions were not included. Recently, Lahne et al. proposed a list of flavor descriptions for American and rye whiskey that does not contain any descriptions of sulfur character [71], probably because the wheel would apply only to matured products that contain low levels of sulfur volatiles. They applied the list to a comparison of bourbons with rye whiskeys [72]. Arnold et al. proposed a list of terminologies for corn whiskey with reference compounds and materials [73]. They defined intensities in two or three concentrations of reference compounds for various attributes and applied them to a terroir of corn new-make whiskeys. As Miller pointed out, these might represent a starting point for a scientific description of American whiskey [29] (p. 57), and it is expected that the attributes of American whisky will be categorized systematically in the future.

Kyraleou et al. investigated the effect of terroir on flavor components in new-make malt whisky [74]. They compared new-make spirits differing by two varieties of barley, two growing environments, and two seasons, showing that environments and seasons had a greater impact on flavor than variety alone. With regard to sulfur compounds, they found that 3-(methylthio) propanal was likely to be one of the key contributors by terroir.

Jack and Fotheringham investigated the relationship between the sensory score for the sulfury attribute and the amount of sulfur compounds [75]. The authors observed that eight typical sulfur volatiles, including DMS, DMDS, DMTS, and thiophenes, showed a high one-to-one association between sensory score and concentration. A weak relationship for the prediction of sensory intensity based on 36 identified sulfur compounds was observed; however, the relationship was strengthened by the addition of MMFDS, suggesting that MMFDS might contribute substantially to a sulfury odor.

Daute et al. evaluated three approaches to access flavor profiles [13], including GC-MS analysis, quantitative descriptive analysis, and Napping sensory analysis. The authors prepared several spirits with different ABV on a laboratory scale, carried out two sensory analyses, and determined the amount of 96 volatile compounds including DMDS and DMTS by GC-MS. It seems that a significant relationship was observed between the concentrations of only two sulfides and the related sensory scores for feinty, cereal, sulfury, and meaty. The authors suggested that cereal character might be derived from the interaction of some volatiles, because no cereal peaks were detected by GC-olfactometry.

As of yet, the threshold of every sulfur volatile found in whisky has not been determined. In addition, the threshold might differ between malt whisky and grain whisky, Scotch and American whiskey, and matured and new-make spirits; moreover, sensory analyses may determine different thresholds due to individual variations among the panelists [60]. Lee et al. determined the thresholds of 16 aroma compounds in a grain whisky matured for 3 years [76], and reported 4 μg/L for DMTS as a sulfur compound. Watt et al. determined the thresholds of six sulfur volatiles in 20% ethanol solution, as mentioned above in the distillation section. Daulby and Wardlaw determined threshold values of DMS, DMDS, DMTS, ethanethiol, methional, and MMFDS of 8.1, 31, 48, 32, 520, and 22 μg/L, respectively [77], although detailed information was not provided. Leppänen et al. examined variations in sulfur volatiles in several commercial products [61], finding that the thresholds differed depending upon the quality. In summary, it may be useful to measure thresholds in a common matrix such as ethanol solution; however, this application may be limited to only new-make spirits.

The sulfury attribute seems to be an unpleasant character; however, there have been several reports of a positive impression at low levels. Harrison et al. pointed out that sulfur volatiles may contribute positively to complexity at low levels [14], although they are unpleasant at high levels. Walker and Hill also indicated that their presence might contribute to the heaviness or body of the final spirit [17]. Moreover, Yomo et al. described that new-make spirits containing considerable amounts of DMTS and dithiapentyl derivatives, derived from brewer’s yeast, might have a full body after maturation [54]. Hence, in the future, the relationship between quality and sulfur volatile levels, through both the determination of thresholds and advances in analytical measurement, might provide a better consistency of product; furthermore, their contribution at a certain level might consequently lead to a higher quality product. Overall, it seems to be hard to demonstrate a relationship between the concentration of flavor compounds and sensory attributes: multiple flavor compounds might contribute to a single attribute, and similarly, a single compound might contribute to multiple attributes with different weights [9].

## 5. Control of Sulfur Compounds in Whisky

The amount of sulfur volatiles in whisky might be controlled either upstream or downstream, thereby decreasing the amounts formed or reducing the levels via some process. In this regard, treatment with activated carbon or charcoal is well known as a downstream solution (pp. 309–310), [29,78,79,80]. This treatment seems to be a non-selective reaction and removes both desirable and unpleasant components in alcoholic beverages. In addition, the type of activated carbon affects the quality of the filtrate. Dauby and Wardlaw focused on the removal of six sulfur volatiles, namely DMS, DMDS, MDTS, ethanethiol, 3-methysulfanylpropanal, and MMFDS, by filtration through several activated carbons, including coal and coconut shell-based activated granular carbons, and evaluated the sulfur intensity using sensory analysis [77]. They observed a notable improvement in organoleptic score for two types of activated carbon, suggesting that these might remove sulfur volatiles significantly. Magee also reported the relationship between the pore size of activated carbon and sulfur components, showing that the pore size and duration of contact affected the content [81].

Charcoal made from sugar maple is used for the production of Tennessee whisky in a method called the Lincoln process. Kerley and Munafo previously investigated the influence of this charcoal filtration [68], showing that in sensory evaluation lower scores for rancid, fatty and roasty were observed for the treated spirits. In terms of sulfur volatile compounds, the amount of both 1-(1,3-thiazol-2-yl) ethenone and DMTS was decreased. Overall, the selection of the charcoal type and filtration condition might need to be established for other whiskies due to the non or low specification of compound removal.

A petrochemical company and a distiller have developed a novel filtration system for the removal of sulfur volatiles. Sugimoto et al. reported that food-quality silver-supported zeolite selectively removed DMS and DMDS in malt whisky [82,83]. In a comparison with activated carbon, the silver zeolite changed none of the major volatiles, including esters and fusel alcohols, while the activated carbon decreased these volatiles [84]. This technology might provide a way to optimize the maturation period, and eventually lead to the control of sulfur volatiles in whisky. Interestingly, a gold nanoparticle supported on silicon dioxide has been reported for the selective removal of DMTS in Japanese sake [85]; as yet, however, this approach does not seem to have been applied to distilled spirits.

Distillers have always had to wait for sulfur volatiles to decrease during maturation because, there have not yet been any technologies for their control. At present, there is fragmented information on the control of sulfur volatiles in malting, fermentation, and distillation, but the integration of these data could result in active control technology. Furthermore, it is expected that useful innovations for the control of volatiles will be developed in the future and will enable distillers to achieve a better command of the optimization of quality. On the other hand, a few researchers have pointed out that trace amounts of some sulfur volatiles contribute positively to quality. However, it may be very difficult to control volatiles to a certain level with our current knowledge and technology. Notably, no scientific reports on whisky blends were found for this review, and so this topic was not covered, but it may be no exaggeration to say that blending is a method associated with optimized whisky quality.

## 6. Conclusions and Looking Forward

Sulfur volatile compounds in whisky have long received attention because they contribute at low levels to quality. Many are formed during malting, fermentation, and distillation, while some are decreased during distillation and maturation. As of yet, their formation pathways have not been elucidated completely, and there have been few experiments on reducing their amounts during upstream processes.

In terms of controlling sulfur volatiles, DMS is reportedly produced by heat during the kilning of malt; thus, the development of this process might lead to a decrease in its formation. In addition, several sulfur volatiles, including hydrogen sulfide, ethanethiol, MTP, MTPA, DMS, MDDS and DMTS are formed by yeast or chemical reactions during fermentation. Therefore, fermentation conditions and novel yeast strains with low production levels of hydrogen sulfide might lead to a decrease in several sulfur volatiles. While Harrison proved that the copper of stills removes sulfur volatiles during distillation [14], a few researchers have proposed that alkyl sulfides may also form during this step; thus, formation and removal may occur simultaneously. Details on the material balance between formation and removal is needed for the integration of these two theories. Lastly, levels of sulfur volatiles decrease during maturation; however, it is a fact that the only distinct method for decreasing sulfur volatiles in conventional whisky making is maturation. Further innovations for the control of sulfur volatiles should be developed. Recently, several accelerated aging technologies have been introduced, in which treatments such as oak chips coupled with high pressure, high temperature, and sonic waves, among others, might shorten the maturation period [5,86]. These sustainable technologies have so far focused on only the extraction of oak flavors and development of color, but both extraction and adsorption are crucial for the final product. Therefore, their combination with other technologies that reduce the amount of sulfur volatiles would be useful to achieve consistency in the future.

Sensory evaluations including flavor wheels have been established systematically and provide useful information on consistency and batch quality. Due to the complex matrix of whisky, the industry has traditionally relied upon sensory evaluation. The relationships between sensory attributes and sulfur components have not been elucidated in full, but this issue may be difficult to resolve because one compound may contribute to several attributes, and multiple compounds might contribute to a single attribute with different weights. Determining compound thresholds might be helpful in this respect, especially as a few researchers have indicated that sulfur compounds at low levels contribute to quality positively. The further development of chemical analyses including novel detectors and in silico techniques might provide solutions in the future.

## Figures and Tables

**Figure 1 molecules-27-01672-f001:**
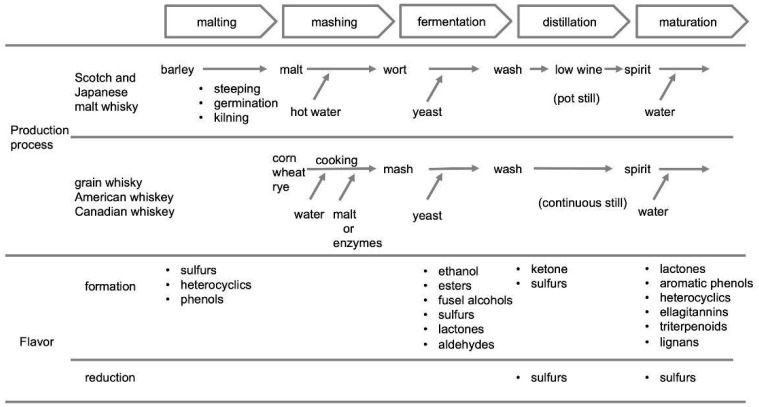
The production processes of Scotch and Japanese malt and grain, and American and Canadian whiskey, showing where sulfur compounds are formed and where they are decreased.

**Figure 2 molecules-27-01672-f002:**
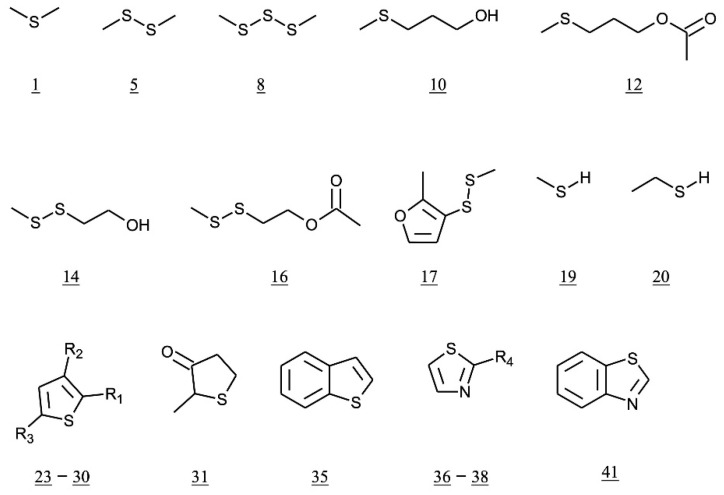
Structures of major sulfur volatile compounds found in whisky. Number of each compound is shown in Table 2, as follows: 23, R_1_ = R_2_ = R_3_ = H; 24, R_1_ = CH_3_, R_2_ = R_3_ = H; 25, R_1_ = R_3_ = CH_3,_ R_2_ = H; 26, R_1_ = CHO, R_2_ = R_3_ = H; 27, R_1_ = R_3_ = H, R_2_ = CHO; 28, R_1_ = CHO, R_2_ = R_3_ = H; 29, R_1_ = CHO, R_2_ = C_2_H_5_, R_3_ = H; 30, R_1_ = CHO, R_2_ = H, R_3_ = CH_3_; 36, R_4_ = H; 37, R_4_ = CH_3_; 38, R_4_ = COCH_3._

**Table 1 molecules-27-01672-t001:** Type of material and distillation method in worldwide whisky.

	Scotch and Japanese Whisky		Irish Whiskey		American Whiskey	Canadian Whiskey
	Malt Whisky	Grain Whisky	Pot Still Whiskey	Grain Whisky		
materials	malted barley	corn, wheat, malted barley	malted barley, barley	corn, wheat, barley	corn, wheat, rye, malted barley	corn, wheat, rye, malted barley, malted rye
distillation	batch	continuous	batch	continuous	continuous	continuous

## Data Availability

Not applicable.

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
