# Peer review of "A Narrative Review of Sulfur Compounds in Whisk(e)y"

_molecules, 2022, doi:10.3390/molecules27051672_

Round 1

Reviewer 1 Report

  1. Some figures should be included
  2. Grammatical corrections need to be corrected
  3. The conclusion should be concise.

Author Response

We thank you for useful comments.

  1. Some figures should be included
    • We have updated all figures.
  2. Grammatical corrections need to be corrected
    • We had  a native speaker proofread our manuscript agian.
  3. The conclusion should be concise.
    • We would like to demonstrate future studies based on the conclusion in the section of "Conclusion and looking forward".  We have shorten this section according to the suggestion so that it would be easy to understand.

Reviewer 2 Report

It is a very interesting review of volatile sulfur compounds in whisky. I found this to be very interesting and well-written. I am convinced that all the information will be important to people interested in whisky production, chemical composition, and quality. 

Secondary comment:

 I think there should be (Scotch and Japanese whisky) in the caption (fig. 1) as well.

Author Response

Dear a reviewer,

  1. I think there should be (Scotch and Japanese whisky) in the caption (fig. 1) as well.
    • Thank you for useful comment.  We have added "Scotch and Japanese malt whisky"  in fig 1 as your suggestion and changed the caption as " The production processes of Scotch and Japanese malt and grain, American and Canadian whiskey, showing where sulfur compounds are formed and where are decreased."

Regards,

Reviewer 3 Report

The authors have done an excellent review on the occurrence of sulfur compounds in whiskeys.

However a few minor typos  are notable and are highlighted as follows:

  1. Rephrase sentence in lines 38,39 and 40 for clarity.
  2. Line 64 describing types of mashing. I think the authors could have been referring either to the steps  involved and not types as described in the text.
  3. Line 184. Kindly add the other thiol  in the text since they are merely three
  4. Table 3. Give the legend at the beneath the table especially for the abbreviation 'nd' used in the table
  5. Line 444. Rewrite as ' 

    With regard to sulfur compounds, they found that 3-(methylthio) propanal was likely to be one of key contributors by terroir.

  6. Line 572. Italicize  the word in silico
  7. Line 538. The author assert that few experiments have been performed is not correct. I suggest they change to few experiments  have been reported in literature.

Author Response

Dear a reviewer,

I really appreciate for useful suggestions.  We have changed our manuscript as your suggestion.

  1. Rephrase sentence in lines 38,39 and 40 for clarity.
    • We have changed this sentense as "starch and proteins are degraded to sugars and amino acids mainly by malt or commercial enzymes during mashing process; ethanol is produced from the sugars by yeast during fermentation; the ethanol is concentrated during distillation; and the final spirit is matured in oak." 
  2. Line 64 describing types of mashing. I think the authors could have been referring either to the steps  involved and not types as described in the text.
    • We have added some references in this sentence as "There are two types of mashing: separation from solid, and whole mash [1,4,10]"
  3. Line 184. Kindly add the other thiol  in the text since they are merely three
    • We have added one more compound, sulfane, as your suggesiton.
  4. Table 3. Give the legend at the beneath the table especially for the abbreviation 'nd' used in the table
    • We have changed nd to "not shown".  Probably the researchers used some extraction method, but unfortunately they did not show it because the report was a technical note which focused on multidimensional GC and several detectors.
  5. Line 444. Rewrite as ' 

    With regard to sulfur compounds, they found that 3-(methylthio) propanal was likely to be one of key contributors by terroir.

    • I really thank you for your suggestion.  We have changed it according to your suggestion as "they found that 3-(methylthio) propanal was likely to be one of key contributors by terroir." (line 441)
  6. Line 572. Italicize  the word in silico
    • We have changed it as your suggestion. (line 564)
  7. Line 538. The author assert that few experiments have been performed is not correct. I suggest they change to few experiments  have been reported in literature.
    • We have changed it as your suggestion, " and there have few experiments on reducing their amounts during upstream processes." (line 533)

Reviewer 4 Report

The reviewed manuscript is about sulfur volatile compounds in whisky. It is quite an interesting paper, but rather for a small group of readers. I have the most objections to language and the way of presenting the results. There are a lot of simplifications and sentences taken out of context which makes the reception difficult. 

I can recommend this text for publication after minor corrections:

- The abstract should be written more precisely:

1.  compare lines 9-10 vs lines 13-14

"Sulfur volatile compounds generated during this process have long attracted interest because they have a negative effect on quality" vs

"It is generally thought that sulfur volatiles contribute to quality positively at low levels, but negatively at high levels."

It is a bit misleading.

2. "More than forty compounds..." - see table 2; there are 39 compounds

3. lines 11-12:  "...but are reduced or decreased during distillation..." - what does it mean? it is incomprehensible.

- lines 66-67: " In addition..." - the sentence is out of context

- many times in the text: formation and reduction of sulfur compounds; - I think it is necessary to clearly explain - it is about reducing the content of sulfur compounds (removal) and not their reduction in the chemical sense.

- line 161 - please add "of sulfur compounds"

- line 168-170: "DMS yields cooked sweet corn, cooked vegetables, and cooked tomato [38,39], and reportedly has a threshold of 5 μg/L in 20% ethanol solution [40]." - it is incomprehensible and confusing

Table 2: what does it mean: "description" - could you be more specific

Figure 2: compound 37 is not drawn correctly

lines 361-362: "multi-dimensional GC coupled..." with?

lines 369-370: "and subsequently detected them in blended whiskies or products" - what kind of products do you mean?

Typos:

line 191 - not described

line 389 - methylsulfanylpropane

line 434 - proposed

References:

[18] - use the journal title abbreviation

[34] - In Proceedings of the Proceedings?

[42] - the article title is given twice

Author Response

Dear a reviewer,

We really thank you for useful comments. 

- The abstract should be written more precisely:

1.  compare lines 9-10 vs lines 13-14

"Sulfur volatile compounds generated during this process have long attracted interest because they have a negative effect on quality" vs

"It is generally thought that sulfur volatiles contribute to quality positively at low levels, but negatively at high levels."

It is a bit misleading.

  • We have changed these sentences as following.  "Their contribution to overall quality depends on their concentration, a positive contribution at low levels, but negative contribution at high levels." (line 13)

2. "More than forty compounds..." - see table 2; there are 39 compounds

  • We have updated table 2.  There are 43 compounds in table 2.  

3. lines 11-12:  "...but are reduced or decreased during distillation..." - what does it mean? it is incomprehensible.

  • We corrected this sentense accurately as your suggestion as "they are formed during malting, fermentation, and distillation, but some may decrease in concentration during distillation and maturation." (line 10-12)

- lines 66-67: " In addition..." - the sentence is out of context

  • We deleted this sentense as your suggestion.  But we described about amino acids in Line 41 and 61.  Proteinases are formed during malting and degrade proteins to amino acids during mashing.  These are necessary for growth of yeast.

- many times in the text: formation and reduction of sulfur compounds; - I think it is necessary to clearly explain - it is about reducing the content of sulfur compounds (removal) and not their reduction in the chemical sense.

  • We have updated in these expressions "formaiton and removal".  For example,
    • Various flavor compounds are formed and also reduced in quantity during these processes [8,9], as shown in Figure 1 (line 44)
    • 2. Formation and removal of sulfur compounds in the whisky production process (line 152)
    •  In this section, we describe how some of these compounds are formed, as well as where their levels are decreased, in the whisky-making processes. (line 160)

- line 161 - please add "of sulfur compounds"

  • We have changed the figure capture as "The production processes of Scotch and Japanese malt and grain, American and Canadian whiskey, showing where sulfur compounds are formed and where are decreased.".

- line 168-170: "DMS yields cooked sweet corn, cooked vegetables, and cooked tomato [38,39], and reportedly has a threshold of 5 μg/L in 20% ethanol solution [40]." - it is incomprehensible and confusing

  • We have changed it as your suggestion, as "DMS is described as “cooked sweet corn”, “cooked vegetables”, and “cooked tomato” [39,40], and its threshold is reported as 5 μg/L in 20% ethanol solution [41] "(line 168)

Table 2: what does it mean: "description" - could you be more specific

  • We have corrected it.  We changed it "odor description" as your suggestion.

Figure 2: compound 37 is not drawn correctly

  • We have updated Figure 2..

lines 361-362: "multi-dimensional GC coupled..." with?

  • We have corrected this sentence as "GC coupled using an electron capture detector (ECD) and a heart-cut technique".

lines 369-370: "and subsequently detected them in blended whiskies or products" - what kind of products do you mean?

  • We have changed "blended whiskies or products" to "13 blended and 3 malt whiskies" more precisely.

Typos:

line 191 - not described

line 389 - methylsulfanylpropane

line 434 - proposed

References:

[18] - use the journal title abbreviation

[34] - In Proceedings of the Proceedings?

[42] - the article title is given twice

  • We really appreciate for bringing it to our attention.  We have corrected them as your indications.